# Epidemiological Profile and Diagnostic Outcomes of Blood Donors Following Hepatitis B Screening at the Largest Blood Bank in the State of Pará, Brazil

**DOI:** 10.3390/idr17060145

**Published:** 2025-11-28

**Authors:** Núbia Caroline Costa de Almeida, Beatriz Monteiro Rodrigues Coelho, Camila Fonseca Barroso, Carlos Eduardo de Melo Amaral, Renata Bezerra Hermes de Castro, Letícia Martins Lamarão, Jacqueline Cortinhas Monteiro, Lucimar Di Paula dos Santos Madeira, Igor Brasil-Costa

**Affiliations:** 1Foundation Center for Hemotherapy and Hematology of Pará, Nucleic Acid Test (NAT) Department, Belém 66033-000, PA, Brazil; 2Evandro Chagas Institute, Health Ministry of Brazil, Ananindeua 67030-000, PA, Brazil; beatrizrodriguesmc2@gmail.com; 3Institute of Biological Sciences, Federal University of Pará, Belém 66075-110, PA, Brazil

**Keywords:** hepatitis B, transfusion, blood donors, follow-up

## Abstract

Background/Objectives: Serological and molecular screening for Hepatitis B virus (HBV) has been essential in reducing the risk of transfusion-transmitted infection, particularly in regions of high endemicity. This retrospective study aimed to analyze the epidemiological profile and laboratory outcomes of 259 blood donors deemed ineligible after initial reactive or inconclusive screening for HBV markers. Methods: Donors were summoned for revaluation at the HEMOPA Foundation, in Belém, Pará, between February 2015 and July 2016. Demographic data, risk factors, and results for HBsAg, anti-HBc, anti-HBs, and HBV DNA obtained at the donation and return time points were collected. Results: The mean age was 37 ± 11.25 years, with a predominance of males (56.8%) and first-time donors (76%). At the return time point, 63.7% presented a profile indicative of resolved HBV infection and 3.5% of active infection, 6.6% were susceptible to HBV infection, and 1.9% presented vaccine-induced HBV immunity. Cases of Occult Hepatitis B Infection (OBI, 0.4%) and Window Period (WP, 0.4%) were also identified. Conclusions: The findings reveal a high prevalence of resolved HBV infection among ineligible donors, particularly first-time donors, and reinforce the importance of combined serological and molecular screening, as well as the need for vaccination and health education strategies for at-risk populations. As a public blood bank located in the Amazon region, we highlight that local epidemiological specificities must be considered in the formulation of public health policies that are sensitive to the regional context.

## 1. Introduction

Hepatitis B virus (HBV) belongs to the *Hepadnaviridae* family and *Orthohepadnavirus* genus and is transmitted through contact with contaminated blood and other bodily fluids. The infection may manifest acutely or progress to a chronic form [1,2]. Hepatitis B is a global public health problem, responsible for high morbidity and mortality rates. In 2022, it was estimated that approximately 254 million people were living with chronic HBV infection, resulting in 1.1 million deaths, mainly from hepatocellular carcinoma and liver cirrhosis [3].

Despite the availability of effective vaccines, it is estimated that more than 1.2 million new infections occur worldwide each year [3]. In Brazil, between 2000 and 2022, a total of 276,646 cases were confirmed, with higher detection rates in the North region compared to the national average. In 2013, for example, the national rate was 8.2 per 100,000 inhabitants, whereas the North region reported 18.8 per 100,000 [4].

Transfusion medicine, or hemotherapy, consists of the transfusion of blood components and blood derivatives for the treatment of different clinical conditions [5]. Considering the risk of blood-borne infections, the Brazilian Ministry of Health Ordinance of Consolidation GM/MS No. 5, dated 28 September 2017, stated that mandatory clinical and laboratory screening is required for all donation candidates [6]. In Brazil, laboratory screening includes testing for the hepatitis B surface antigen (HBsAg) and antibody to hepatitis B core antigen (anti-HBc) and the Nucleic Acid Test (NAT) [6]. Combined testing for HBsAg and anti-HBc at different stages of infection increases the probability of detecting viral presence [7].

However, anti-HBc reactivity is a major reason for the disposal of donations in blood centers, directly impacting blood supply and the operational costs of transfusion services [8,9]. The introduction of the NAT has reduced the residual risk of transmission by detecting viral DNA during the WP [10].

In addition to initial screening, monitoring donors with reactive or inconclusive results is essential to confirm or rule out infection and to provide appropriate support [11]. By reducing diagnostic uncertainty, this process also helps to minimize the emotional impact associated with reporting results, since the possibility of infection and donation deferral may generate anxiety and stress. In this context, structured follow-up strategies, such as clear counseling and psychological support, strengthen the confidence of these donors and transfusion safety [12].

However, in Brazil, few studies have described the epidemiological profile and serological outcomes of these donors after the second blood collection, particularly in regions of high endemicity such as the North region. In this context, the present study aimed to analyze the epidemiological profile and the serological and molecular results of ineligible donors attended at a blood center in Belém, Pará.

## 2. Materials and Methods

### 2.1. Study Population and Ethical Aspects

This retrospective study was conducted at the Hemotherapy and Hematology Center Foundation of Pará (HEMOPA). Data from voluntary donors (*n* = 93,891) received between February 2015 and July 2016 were obtained from the HEMOPA records. The whole blood obtained from participants was processed and subjected to transfusion transmissible disease screening.

During this period, 1086 donors (1.156%) presented at least one reactive or inconclusive marker for HBV in the laboratory screening. According to the internal protocol of the HEMOPA Foundation, donors considered ineligible during screening were recalled to the blood center for collection of a second sample in order to confirm the reactivity initially observed. Upon return, 259 donors were included in the study, and a semi-structured questionnaire was administered, covering the following variables: gender, age, marital status, usual occupation, educational level, HBV vaccination status, and risk factors for HBV infection.

The study was approved by the Research Ethics Committee of the Santa Casa de Misericórdia do Pará Foundation (approval no. 961.726, approved on 24 February 2015). Participants were invited and informed about the study objectives, and their acceptance was formalized by signing the Free and Informed Consent Form. All stages of the research were conducted in accordance with the guidelines of Resolution 466/2012.

### 2.2. Serological Analysis

Serum samples were subjected to chemiluminescent serological assays for the detection of HBsAg (Architect HBsAg Qualitative II, Abbott Diagnostics, Abbott Park, IL, USA), total anti-HBc antibodies (Architect Anti-HBc II, Abbott Diagnostics, Abbott Park, IL, USA), and hepatitis B surface antibody (anti-HBs) (Architect Anti-HBs Quantitative, Abbott Diagnostics, Abbott Park, IL, USA). All tests were performed on the Architect^®^ i2000 automated chemiluminescence platform (Abbott^®^ Diagnostics, Abbott Park, IL, USA), according to the manufacturer’s recommendations, with the aim of confirming the initial reactivity detected during serological screening at the donation time point. In accordance with the official serological screening algorithm established by the Brazilian Ministry of Health (Consolidation Ordinance No. 5, 2017), anti-HBc testing was performed using total antibody detection (IgG + IgM), as established by the Ministry’s official blood screening protocol for transfusion safety [6].

For the HBsAg and anti-HBc assays, the results are expressed in relative light units (RLU) with a cut-off (CO) of 1.0. Samples with RLU/CO values below 0.8 are considered non-reactive; those with values between 0.8 and 1.2 are considered inconclusive, while samples with RLU/CO values above 1.2 are reactive. The anti-HBs assay is quantitative, with results expressed in mIU/mL. Samples with concentrations below 8 mIU/mL are non-reactive; values between 8 and 12 mIU/mL are considered inconclusive, and results above 12 mIU/mL are reactive.

Inconclusive or weakly reactive samples were retested three consecutive times using the same assay. Samples that remained reactive after repeat testing were subsequently submitted to a confirmatory molecular test (Artus HBV QS-RGQ, QIAGEN, Hilden, Germany). This second, independent method was used specifically to validate doubtful serological results, ensuring compliance with national screening standards and providing analytical robustness to the final classification of the samples. This two-step procedure ensured analytical reliability.

### 2.3. Molecular Analysis

HBV DNA was detected using the HIV/HCV/HBV NAT BioManguinhos assay (Fiocruz, Rio de Janeiro, Brazil), commercial NAT kit based on real-time PCR methodology. The method was used for molecular screening of donors, following the manufacturer’s protocol (Fiocruz, Rio de Janeiro, Brazil), and was performed on the 7500 Real-Time PCR System thermocycler (Thermo Fisher Scientific, Waltham, MA, USA).

### 2.4. Interpretation of Results

The interpretation of the serological and molecular HBV markers was based on the results obtained at the return time point. Donors were classified into specific infection or immunity profiles according to the combination of HBsAg, total anti-HBc, anti-HBs, and HBV DNA results, as detailed in Table 1.

### 2.5. Statistical Analysis

The data were organized and analyzed using Microsoft Excel^®^ 2016 MSO (Version 2508, Build 16.0.19127.20302, Microsoft Corp., Redmond, WA, USA) and RStudio^®^ (version 2025.05.0, R Core Team, Vienna, Austria). Descriptive analysis included calculations of absolute and relative frequencies, mean, standard deviation, and minimum and maximum values, as appropriate for each variable.

To compare the results of serological and molecular markers at the two collection time points (donation and return), the McNemar test was applied. The proportions test was used to verify differences between serological profiles defined based on the combination of markers. Associations between reactive markers and demographic variables or risk factors were evaluated using the chi-square test and Fisher’s exact test, or its Monte Carlo simulation version when appropriate.

To assess linear trends between ordered age groups and the prevalence of the anti-HBc marker, the Cochran-Armitage trend test was applied. Additionally, a Locally Estimated Scatterplot Smoothing (LOESS) regression line was used to illustrate non-linear trends in the distribution of data in graphs.

The significance level of 5% (*p* < 0.05) was adopted for all statistical analyses.

## 3. Results

### 3.1. Donors Profile

This study included 259 blood donors who returned to HEMOPA for collection of a second sample to confirm serological ineligibility. Of the total, 56.8% were male and 43.2% were female, with a male-to-female ratio of 1.3:1. The mean age of participants was 37 ± 11.25 years (ranging from 18 to 63 years). When stratified by age group, the highest frequency was observed in the 36–45 age group (29.7%), and the lowest frequency was among individuals aged 60 years or older (3.1%) (Table 2). Regarding donor type, the majority of participants (76%) were first-time donors, 20.5% had donated previously, and 3.5% did not report this information. The interval between the first and second samples included in this study ranged from 7 to 389 days (mean: 78.6; median: 37.5), and most donors (69.5%) returned within three months after the first donation.

Regarding usual occupation, most donors (37.1%) worked in general services and manual labor positions, including construction, security, transportation, and other operational or support roles (such as blacksmith, car washer, cargo handler, attendant, merchant, or kitchen assistant). The analysis of educational level showed that 54.8% of participants had secondary education, whether incomplete or complete, while most were married (57.5%) (Table 3).

Regarding HBV vaccination, 45.6% of donors did not know whether they had been vaccinated, 32% reported having been vaccinated, 21.6% stated that they had not been vaccinated, and 0.8% did not respond. Concerning risk factors for HBV infection, 50.6% of donors reported having undergone invasive dental procedures, 50.6% stated they never used condoms, 50.2% had a history of previous surgery, 40.5% reported sharing manicure/pedicure instruments, 22% shared razors or blades, 17% had tattoos, 10.4% reported illicit drug use (including marijuana, cocaine, “glue,” and ecstasy), and 6.2% reported a history of blood transfusion (Table 4).

### 3.2. Frequency of Markers at Donation and Return

At the donation time point, HBsAg was reactive in 4.6% (12/259) of individuals, a percentage that remained unchanged in the return time point (4.6%; 12/259) (Figure 1). However, the overall distribution of categories (reactive, non-reactive, and inconclusive) showed a statistically significant variation between the two time points (*p* = 0.01173). Most donors remained non-reactive in both samples. A small number of individuals showed changes in their results: one donor who was initially reactive became non-reactive at return, and one donor with an inconclusive result at donation tested reactive at the second assessment. These individual shifts account for the statistical difference observed between the two time points.

Total anti-HBc was reactive in 76.4% (198/259) of donors at the initial donation, with a slight reduction to 74.5% (193/259) at return (Figure 2). Although the overall frequency of anti-HBc positivity remained similar between the two samples (76.4% vs. 74.5%), small individual shifts between reactive, inconclusive, and non-reactive results were detected (*p* = 0.0089), possibly reflecting transient immune or analytical variation.

HBV DNA was initially detected in 3.9% (10/259) of donors, increasing to 4.2% (11/259) on return time point. Statistical tests could not be applied to these data. Anti-HBs was evaluated only on return time point, with reactivity in 72.6% (188/259) of individuals.

Analysis of the association between HBsAg positivity and detectable HBV DNA on return showed a statistically significant correlation (*p* < 0.0001), with an estimated odds ratio of 307.28 (95% confidence interval: 43.26–4480.66). Among HBsAg-positive individuals, 75% had detectable HBV DNA. In contrast, only 0.81% of HBsAg-negative individuals presented viral DNA, corresponding to cases compatible with the WP and OBI. Figure 3 shows the distribution of Anti-HBc, HBsAg, and NAT results at donation and return. A small decrease was observed in inconclusive Anti-HBc results (36 to 30) and an increase in non-reactive HBsAg results (227 to 237), while NAT results remained virtually unchanged (248 to 247 undetectable). Overall, the results indicate analytical stability between donation and return time points.

### 3.3. Serological and Infectious Profiles at Return

Based on the combination of serological and molecular markers (HBsAg, total anti-HBc, anti-HBs, and HBV DNA) obtained at the return time point, it was possible to classify the 259 donors into different profiles of exposure, immunity, or HBV infection.

The majority of individuals presented a profile compatible with resolved HBV infection, evidenced by the concomitant presence of anti-HBc and anti-HBs, without detection of HBsAg or viral DNA, corresponding to 63.7% (165/259). Donors classified as susceptible to HBV infection, characterized by the absence of all markers, totaled 6.6% (17/259).

Cases compatible with active HBV infection, defined by the simultaneous presence of HBsAg, anti-HBc, and detectable HBV DNA, corresponded to 3.5% (9/259). All nine donors classified with active infection were HBsAg-reactive and had detectable HBV DNA at the return time point. These same individuals had already shown HBsAg positivity at the donation time point, which was confirmed in the follow-up samples. Among them, eight had detectable HBV DNA in both donation and return samples, whereas one donor’s initial sample was not tested by the NAT but remained HBsAg-reactive at both time points.

The serological and molecular pattern indicative of OBI (anti-HBc positive and detectable DNA, with HBsAg negative) was observed in 0.4% (1/259). Another 0.4% (1/259) presented a profile suggestive of the WP, with isolated detection of HBV DNA, without reactive antibodies or antigen.

Individuals with vaccine-induced HBV immunity, with isolated reactive anti-HBs, totaled 1.9% (5/259). The isolated anti-HBc profile was identified in 6.2% (16/259) of donors. Finally, inconclusive serological results were categorized as possible false positives, corresponding to 17.4% (45/259). The interpretation criteria for these profiles are detailed in Table 1. Table 5 details the absolute numbers and percentages of each identified serological and molecular profile.

### 3.4. Association Between Markers and Demographic Variables/Risk Factors

Potential associations were evaluated between the positivity of serological and molecular markers at return and demographic variables (gender, age group, educational level, usual occupation, and marital status), as well as risk factors related to HBV infection.

Regarding HBsAg positivity, no statistically significant associations were observed with any of the analyzed demographic variables or with donor type. Similarly, risk factor variables such as HBV vaccination (*p* = 0.8598), invasive dental procedure (*p* = 1.000), condom use (*p* = 0.9109), use of shared manicure/pedicure equipment, shared razors or blades, tattoos, blood transfusion history, and history of previous surgery, showed no significant relationship with the presence of the marker. However, a statistically significant association was identified between HBsAg positivity and reported illicit drug use (*p* = 0.00278).

Anti-HBc positivity showed a statistically significant in association with donor type (*p* < 0.0001), being markedly more frequent among first-time donors compared with repeat donors. In contrast, no significant associations were observed with other demographic variables. None of the analyzed risk factors were significantly related to anti-HBc positivity, although sharing manicure/pedicure equipment showed a trend toward significance (*p* = 0.05739), suggesting a possible behavioral influence that warrants further investigation.

Analysis of anti-HBc prevalence in different age groups showed that, as age increased, the proportion of donors with reactive results also rose (*p* = 0.01948), suggesting a higher frequency of exposure among older donors. However, a slight decline was observed among donors aged over 60 years, which may be influenced by the small sample size in this group (*n* = 6). Figure 4 presents the percentage distribution of prevalence by age group.

For anti-HBs, a statistically significant association was observed with donor type (*p* < 0.0001), with higher positivity among first-time donors. Conversely, no significant associations were found with other demographic variables or risk factors evaluated, except for the use of shared manicure/pedicure equipment, which showed a trend toward association (*p* = 0.05369). Among donors with detectable HBV DNA, a significant association was observed with gender, being more frequent in males (*p* = 0.02617), as also with reported illicit drug use (*p* = 0.02003). No statistically significant associations were observed with donor type or with other evaluated demographic variables and risk factors.

### 3.5. Association Between Serological Profiles and Demographic Variables/Risk Factors

Donor type was the only variable showing a statistically significant association with serological profiles (*p* < 0.0001), with a higher proportion of resolved previous infections among first-time donors. No statistically significant association was found between gender and serological profiles, although a male predominance was observed in most groups. For other demographic variables, such as age group, educational level, usual occupation, and marital status, no significant associations were observed, indicating that serological profiles were distributed relatively homogeneously across these groups.

Regarding risk factors, HBV vaccination also showed no significant association with the identified profiles (*p* = 0.8236), nor did the use of shared manicure/pedicure equipment (*p* = 0.1221). Additionally, the analysis of the association between gender and illicit drug use revealed a statistically significant relationship (*p* = 0.0003). The estimated odds ratio was 6.84 (95% CI: 1.99–36.50), indicating a higher likelihood of illicit drug use among male donors. Additionally, the same pattern of non-significance was observed for the remaining variables analyzed.

### 3.6. Special Cases: OBI and WP

Two cases with serological profiles considered atypical, compatible with the WP and OBI, were identified. Table 6 presents the main characteristics of these donors, including demographic data, reported risk factors, and laboratory results.

## 4. Discussion

The combined use of serological and molecular markers is essential for HBV screening in blood banks, contributing to the reduction in transfusion-transmitted infection risk [10]. In the present study, conducted with donors deemed ineligible at a public blood center in Pará, a prevalence of 1.156% positivity for at least one HBV marker was observed, resulting in the discard of 1086 blood bags. This percentage is similar to that reported in Brazilian states such as Amapá (1.06%) and Goiás (1.29%) [13,14] but lower than the values documented in African countries such as Ethiopia (7.88%) and Somalia (9.7%) [15,16].

The majority of donors were male (56.8%), with a mean age of 37 years and a predominance of individuals aged 36–45 years. This pattern is consistent with findings from other studies, and the observed gender difference may be related to screening criteria that temporarily defer or permanently exclude women from donation, such as minimum body weight, pregnancy, and hemoglobin levels [14,17,18].

Regarding donor type, the majority of participants (76%) were first-time donors, while 20.5% had previously donated blood. A significant association was observed between donor type and the presence of anti-HBc, anti-HBs, and the different serological profiles, all of which were more frequent among first-time donors. These findings indicate greater previous exposure to HBV in this group, which has not yet undergone continuous screening over time, potentially increasing the residual risk of infection. Similar results have been reported in other studies [19,20,21]. Such evidence reinforces the importance of stimulating regular donor loyalty, a key strategy to ensure greater safety in the blood donation process.

Almost half (45.6%) of the donors did not know whether they had been vaccinated against HBV, while 32% reported having received the vaccine. However, only 1.9% showed a serological profile compatible with vaccine-induced immunity. Among these, most donors were unable to accurately report their vaccination status, despite showing laboratory evidence of immunity. This finding may reflect both the decline in anti-HBs antibody levels over time and failures in the vaccination schedule, such as incomplete administration of the three doses or non-compliance with the recommended intervals. Some donors who reported being vaccinated may not have maintained sufficient anti-HBs levels, possibly due to these same factors, as described in other studies [22,23]. These findings reinforce the need for screening and vaccination strategies in exposed populations, particularly in regions of high endemicity such as the Amazon [3,24,25]. In this context, it is important to recall that the Brazilian National Immunization Program recommends the hepatitis B vaccine in a three-dose schedule for susceptible individuals of all ages, which achieves protective immune responses in over 90% of healthy recipients and remains a key component in preventing HBV infection [7,23].

Exposure to risk factors was common among blood donors called for reevaluation after inconclusive or reactive screening, emphasizing invasive dental procedure, unprotected sexual practices, and shared manicure/pedicure equipment. Such practices favor HBV transmission, particularly in contexts of inadequate sterilization and risky sexual behavior [26,27,28].

Illicit drug use was the only factor that showed a statistically significant association with both HBsAg and HBV DNA, even when the reported substances were non-injectable (marijuana, cocaine, “glue,” and ecstasy). This finding reinforces the vulnerability of this group, which is frequently associated with risk behaviors, such as unprotected sexual practices and multiple partners, factors that potentiate exposure to HBV [29]. Previous studies had already reported this association, suggesting that even among non-injectable drug users, especially long-term users, the risk may be related to parallel risk behaviors or indirect contact with contaminated blood, such as in situations where the individual tries injectable drugs for the first time in a group [29,30,31]. These findings highlight the importance of strengthening prevention and screening measures, as well as implementing targeted health education strategies aimed at populations with greater vulnerability, such as illicit drug users, to help reduce HBV transmission in endemic regions.

Viral DNA detection was more frequent among male donors, possibly due to behavioral or usual occupational differences, as reported in other studies [32,33]. Although most donors in the study were male (56.8%), this predominance likely reflects the general donor population profile rather than a specific association with HBV markers. Therefore, this proportion should be considered when interpreting the overall distribution of results. Therefore, the interpretation should take this initial proportion into account to avoid biased conclusions.

The isolated analysis of markers showed a high positivity of anti-HBc (>70%), higher than that observed in regions such as Caxias do Sul (50.3%) and Sergipe (28.3%) but consistent with data from endemic areas of the Amazon, such as Redenção, Pará (75.4%) [34,35,36]. A rising trend in positivity with increasing age was also identified, reflecting cumulative lifetime exposure, a finding similar to that reported in Ethiopia [37]. However, the slight decline observed among donors over 60 years old may be related to the small sample size in this group (n = 6).

The frequency of HBsAg remained at 4.6%, exceeding the rates reported in studies from Yemen (2.4%) and Ethiopia (4.07%) [38,39]. The detection of viral DNA in 4.2% of donors on return, including cases of OBI and the WP, reinforces the importance of the NAT in reducing the diagnostic WP and increasing transfusion safety [10].

Rare cases were identified, such as OBI (0.4%) and the WP (0.4%), situations that are difficult to detect serologically but were captured by the NAT. The frequency of OBI observed was similar to that reported in São Paulo (0.6%) and Vietnam (0.3%) [40,41]. Regarding the WP, a study in the North region showed that, after the implementation of NAT-HBV, there was an important reduction in residual risk, from 1:144.92 to 1:294.11 donations [10]. The prevalence of active infection (3.5%) in the present study was higher than that reported in São Paulo (0.024%) and comparable to that found in Saudi Arabia (3.24%) [40,42], reinforcing the need for the adoption of the NAT as a complementary method, particularly in areas of higher endemicity.

In 6.2% of donors, isolated anti-HBc was identified without evidence of viral DNA. This profile may reflect a previous HBV infection with subsequent low levels of anti-HBs resulting from the loss of the ability to produce these antibodies due to immunological senescence, since anti-HBc antibodies are known to persist longer than anti-HBs after infection resolution [43]. Alternatively, nonspecific reactivity cannot be completely ruled out. Given the limitations of anti-HBc assays and the risk of false-positive results, it is recommended to confirm these cases using different methodologies and to maintain clear communication with the donor [44].

The majority of donors presented a profile compatible with resolved HBV infection (63.7%), characterized by positivity of both anti-HBc and anti-HBs. This proportion was higher than that observed in other countries, such as China (5.9%) and Bangladesh (10.6%) [45,46], a result that may be explained by the study design, which included only donors already deemed ineligible and called for reassessment. Few studies have specifically focused on this group, underscoring the originality of these findings.

Approximately 6.6% of donors exhibited a profile susceptible to HBV, as they did not present any serological or molecular markers upon return. Among them, about one quarter reported having been vaccinated, while nearly half were unable to recall their vaccination status. These findings suggest that self-reported vaccination may not always correspond to serological immunity, possibly due to incomplete vaccination schedules, waning antibody levels over time, or inaccurate self-reporting of vaccination status [22,23]. These cases probably correspond to initially false-reactive screenings, a phenomenon already described in the literature that may negatively affect the donor’s experience when recalled for additional testing without confirmation of infection [47].

Another important finding was the considerable proportion of possible false-positive serological profiles (17.4%). Most of these individuals showed weak or inconsistent reactivity for anti-HBc and/or anti-HBs, with undetectable HBV DNA. Although these findings may reflect analytical variability or nonspecific reactions, it is also plausible that some donors had resolved infections with very low residual antibody levels, close to the detection limit of the assays. These results may be related to the specific characteristics of the laboratory assays, which in blood banks typically have greater sensitivity than specificity, generating results such as cross-reactions and the presence of heterophilic antibodies, as well as individual variations in immune response [48]. Technical aspects, such as processing interferences, may also contribute to this scenario [49]. Such occurrences tend to be more frequent in regions with high prevalence of previous exposure to HBV, which makes donor confirmation and follow-up more challenging. These findings underscore the importance of well-structured confirmatory protocols and careful communication to minimize the emotional impact of inconclusive results.

Finally, weak HBsAg positivity was observed in three donors, with low reactivity index values (URL/CO) on return and absence of detectable viral DNA, indicating possible cross-reactivity or false-positive, as previously reported [49,50]. These cases reinforce that low-reactivity results should be confirmed by neutralization assays [51,52]. Despite the advances provided by the NAT, the combination of serological and molecular methods still represents the safest strategy for donor screening.

This study has limitations, such as its cross-sectional design and the lack of diagnostic confirmatory in certain profiles. Nevertheless, it provides original and relevant information on donors already deemed ineligible donors in the North region, contributing to the improvement of screening, vaccination, and transfusion safety strategies.

## 5. Conclusions

The results of this study show the complexity of hepatitis B screening among blood donors called for re-evaluation in a highly endemic region such as Pará. A predominance of first-time donors was observed, a group more frequently associated with serological markers for HBV, which reinforces the importance of loyalty to increase transfusion safety.

The high proportion of resolved HBV infection, along with the identification of rare cases such as OBI and the WP, reinforces the importance of integrating serological and molecular methods to increase diagnostic sensitivity and reduce transfusion risk. Although the frequency of HBV DNA was low, the presence of the virus in seronegative individuals highlights the value of the NAT as an essential complementary tool. The association between some viral markers and behavioral risk factors, such as illicit drug use, emphasizes the need to strengthen educational actions and screening and expand vaccination coverage among high-exposure groups. The substantial proportion of inconclusive profiles reinforces the importance of well-structured confirmatory protocols and clear, supportive communication with donors to minimize emotional impact and maintain the link with blood therapy services.

Thus, the presented data contribute to a deeper understanding of the serological and epidemiological profile of these donors already deemed ineligible donors, providing important insights for improving screening practices, epidemiological surveillance, and transfusion safety in the North Region. More comprehensive studies with longitudinal follow-up are recommended to better understand the clinical and immunological outcomes of these individuals.

## Figures and Tables

**Figure 1 idr-17-00145-f001:**
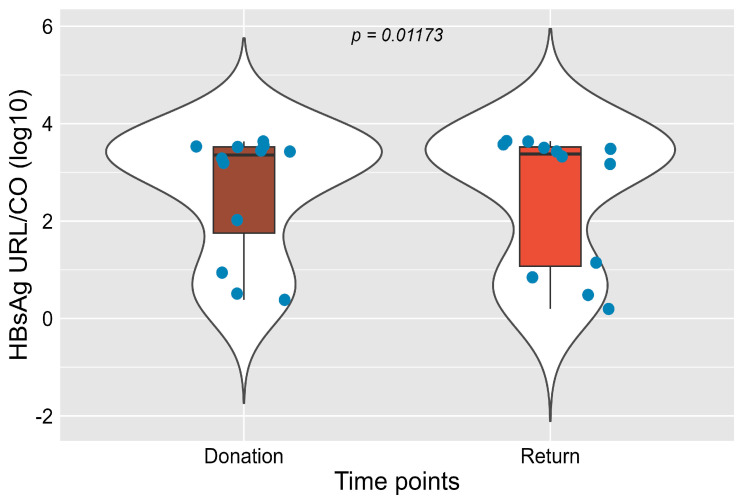
Distribution of HBsAg RLU/CO (log_10_) values at donation and return time points. A significant difference was observed between the two samples (*p* = 0.01173). Blue dots represent individual sample values.

**Figure 2 idr-17-00145-f002:**
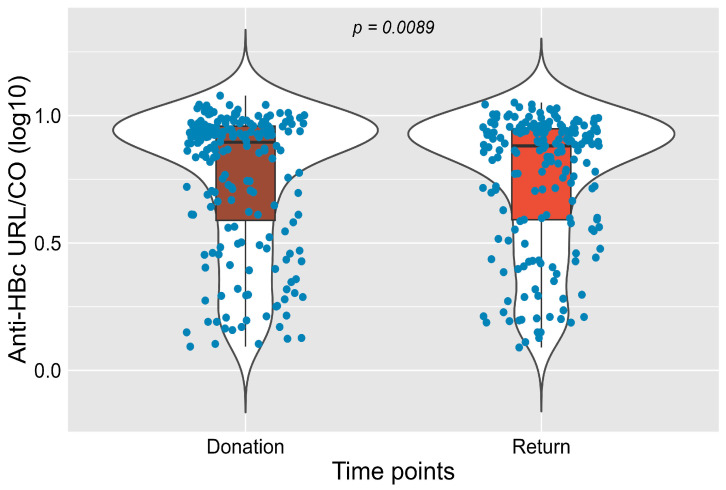
Distribution of total Anti-HBc reactivity (RLU/CO, log_10_) values at donation and return time points (*p* = 0.0089). Blue dots represent individual sample values.

**Figure 3 idr-17-00145-f003:**
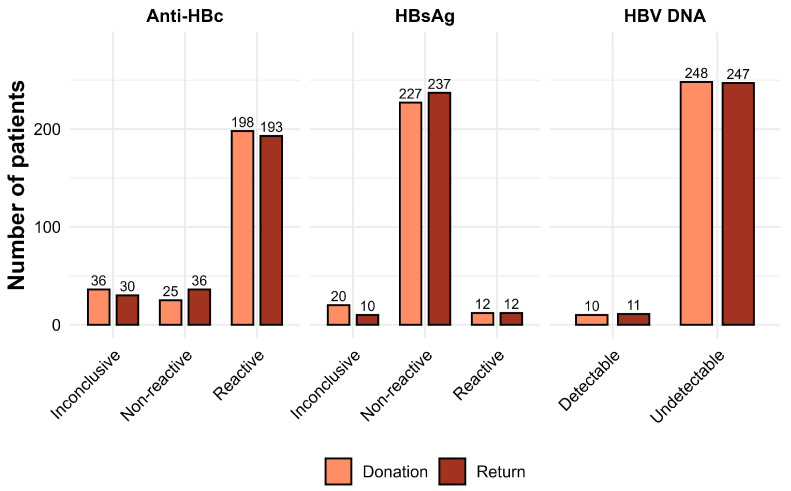
Distribution of individual results for the Anti-HBc, HBsAg, and NAT markers at donation and return time points. Bars show the number of donors classified as inconclusive, non-reactive, reactive (or detectable/undetectable for NAT) for each marker. Legend: One NAT at donation time point and another on the return time point were not performed due to sample depletion, and therefore were not included in the figure.

**Figure 4 idr-17-00145-f004:**
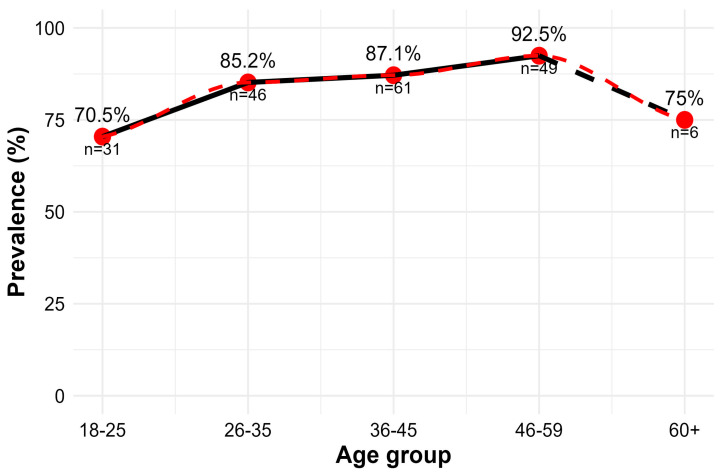
Trend in anti-HBc prevalence among blood donors by age group. The red dashed line represents the smoothed prevalence trend by age group, estimated using the LOESS method. Mean ± SD values of age within each group were: 18–25 (21.9 ± 2.47), 26–35 (30.8 ± 2.52), 36–45 (40.3 ± 2.93), 46–59 (50.7 ± 3.20), and 60+ (61.3 ± 1.51). The dashed (black) line between 46–59 and 60+ age groups indicates that the latter has a smaller sample size (*n* = 6), and therefore the connection between the points is illustrated differently.

**Table 1 idr-17-00145-t001:** Interpretation criteria for HBV serological and molecular profiles.

HBsAg	Anti-HBc	Anti-HBs	DNA	Interpretation/Classification
−	−	+	+	WP
+	+	−	+	Active infection
−	+	inc.	+	OBI
−	+	+	−	Resolved infection
−	−	+	−	Vaccinated against HBV
−	−	−	−	Susceptible
−/+/inc.	−/+/inc.	−/+/inc.	−	Possible false positives
−	+	−	−	Isolated anti-HBc

Legend: (−) non-reactive; (+) reactive; (inc.) inconclusive.

**Table 2 idr-17-00145-t002:** Distribution of gender and age group among the blood donor population.

Variables	N. (%)
Gender	
Female	112 (43.2)
Male	147 (56.8)
Age group (years)	
18–25	48 (18.5)
26–35	67 (25.9)
36–45	77 (29.7)
46–59	59 (22.8)
≥60	8 (3.1)

**Table 3 idr-17-00145-t003:** Demographic profile of blood donors.

Variables	N. (%)
Usual occupation	
General services and manual labor ^1^	96 (37.1)
Unemployed/Informal economy ^2^	84 (32.4)
Administrative/professional sector ^3^	53 (20.5)
Unknown	26 (10)
Educational level	
Primary education (incomplete or complete)	54 (20.8)
Secondary education (incomplete or complete)	142 (54.8)
Higher education (incomplete or complete)	60 (23.2)
Unknown	3 (1.2)
Marital status	
Married	149 (57.5)
Separated/Divorced/Widowed	14 (5.4)
Single	85 (32.8)
Unknown	11 (4.3)

^1^—includes workers in construction, security, transportation (drivers), and operational or support roles. ^2^—includes homemakers, unemployed individuals, retirees, self-employed workers, students and others engaged in informal or non-registered economic activities (including those classified as “others”). ^3^—includes public employees, health professionals, administrative workers, and those in beauty or sewing sectors.

**Table 4 idr-17-00145-t004:** Reported behavioral and clinical risk factors among blood donors (N = 259).

Risk Factors	Yes (%)	No	Sometimes	Always	Never	Ignored
Invasive Dental Procedure	131 (50.6)	128 (49.4)	-	-	-	-
Use Condoms	-	-	56 (21.6)	41 (15.8)	131 (50.6)	31 (12)
Shared Manicure/Pedicure Equipment	105 (40.5)	93 (35.9)	61 (23.6)	-	-	-
Shared Razors or Blades	57 (22)	136 (52.5)	65 (25.1)	-	-	1 (0.4)
Tattoos	44 (17)	209 (80.7)	-	-	-	6 (2.3)
Illicit Drug Use	27 (10.4)	227 (87.7)	-	-	-	5 (1.9)
History of Blood Transfusion	16 (6.2)	240 (92.7)	-	-	-	3 (1.1)
History of Surgery	130 (50.2)	128 (49.4)	-	-	-	1 (0.4)

**Table 5 idr-17-00145-t005:** Distribution of serological and molecular profiles of hepatitis B among evaluated donors.

HBsAg	Anti-HBc	Anti-HBs	DNA	HBV Status	N. (%)	IC95%
−	−	+	+	WP	1 (0.4)	0–2.1%
+	+	−	+	Active infection	9 (3.5)	1.6–6.5%
−	+	inc.	+	OBI	1 (0.4)	0–2.1%
−	+	+	−	Resolved infection	165 (63.7)	57.5–69.6%
−	−	+	−	Vaccinated against HBV	5 (1.9)	0.6–4.4%
−	−	−	−	Susceptible	17 (6.5)	3.9–10.3%
−/+ */inc.	−/+/inc.	−/+/inc.	−	Possible false positives	45 (17.4)	13–22.5%
−	+	−	−	Isolated anti-HBc	16 (6.2)	3.9–9.8%

(−) non-reactive; (+) reactive; (inc.) inconclusive; (*) samples with low reactivity (RLU/CO < 10 or <4). Among the possible false-positive cases, one donor presented HBsAg < 10 and two donors presented HBsAg < 4.

**Table 6 idr-17-00145-t006:** Epidemiological and laboratory profiles of cases classified as WP and OBI.

Variable	Profiles	
	WP	OBI
Gender	Male	Male
Age (Years)	21	47
Usual Occupation	Student	General services
Educational level	Complete secondary	Literate
Marital status	Single	Married
Risk factors		
HBV vaccination	Unknown	Yes (~5 years)
Invasive dental procedure	No	Yes
Condom use	Always	Sometimes
Shared manicure/pedicure equipment	No	Sometimes
Shared razors or blades	No	Sometimes
Illicit drug use	No	No
Tattoo	Yes	No
Blood transfusion history	No	Yes
History of previous surgery	No	Yes
Donation type	Repeat donor	First-time donor
Donation	HBsAg−/Anti-HBc−/DNA+	HBsAg−/Anti-HBc+/DNA+
Return	HBsAg−/Anti-HBc−/Anti-HBs+/ DNA+	HBsAg−/Anti-HBc+/Anti-HBs inc./DNA+

## Data Availability

All relevant data from this study are available from the corresponding author.

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
