# Peer review of "Epidemiological Profile and Diagnostic Outcomes of Blood Donors Following Hepatitis B Screening at the Largest Blood Bank in the State of Pará, Brazil"

_2036-7449, 2025, doi:10.3390/idr17060145_

Round 1
Reviewer 1 Report
Comments and Suggestions for Authors
General comments :
The series studied is "homogeneous" (if one may say so) in that all donors selected for the study were declared "ineligible" due to specific results ("reactive or inconclusive" results) when screened for hepatitis B markers.
The first question that comes to mind is: "Why publish results in 2025 from samples that date back to 2015?" Why not study more recent samples? Wouldn't studying the current epidemiological profile have provided more up-to-date and useful information, especially for donor selection?
On a technical level, one may ask: why not have used separate IgG and IgM tests for anti-HBc? Why was there no 2nd method (an alternative test) to analyze the "dubtful" samples? How are the results of the analyses validated?
The iconography is a little too abundant:
- Table 2 is very detailed in relation to its informative contribution;
- Figures 1 and 2 would deserve some further explanation;
- Figure 3 could be summarized;
- the presentation of Table 3 would benefit from being simplified (see comment related to Line 220)
- Figure 4: while (we know that) the evolution of anti-HBc with age, here between [18 and 59] years of age, is increasing, we do not understand the decrease in anti-HBc observed beyond the age of 60 (too few observations? n=6).
The use of "professional jargon" can be dangerous. See explanations below, in the comments on line 235 and line 260. In summary: for the reader, it is a bit "confusing" to read in the same sentence that there is a correlation between one given parameter (whatever it is) with the "frequency of donations" in "first-time" donors.
The bibliography is well stocked, diversified and well targeted on the subject.
The discussion is very pleasant to read: it systematically takes up the various results obtained, comments on them and places them in a context and with appropriate bibliographical references. Observations that raise questions are most often followed by an answer or hypothesis in the following lines.
This study is not intrinsically original, but it has the merit of presenting with great realism the difficulties encountered in the field by blood bank laboratories with regard to the results of diagnostic tests for the different phases of hepatitis B. These are important analysis results since they play a major role in the decisions on which transfusion safety depends.
To the authors
Specific comments:
Line 85 : “During this period, 1,086 donors presented…” Why not put directly here the percentage that this represents, that is to say: 1.156%. This interesting notion appears only in line 278 in the discussion.
Line 120 : It is not known how the test result is "validated". Is there an algorithm that recommends starting the analysis 2 or 3 times? or who recommends using another test to validate a weakly reactive or inconclusive result, ...
Line 151 : How long did it take between the first and second sampling? Was this delay more or less the same for all retested donors?
Line 154 : The age range for donors is 16 to 63 years. Is 63 years old the maximum age allowed to donate blood? (in other countries, it's 65 years old).
Line 157 : The majority of donors in the series studied are "first time donors" (76%). In some countries, at the time of AIDS in the 1980s and later in the 1990s, when hepatitis C tests were available, there was a rather peculiar phenomenon: people came forward to donate blood, but in reality, their motivation for going to the blood bank was to get tested. Could the same phenomenon be one of the possible explanations ? Or, is the frequency of hepatitis B in the general population so important?
Line 157 : Have you had access to the serological results (Test Ratio) characterizing the first donation?7
Line 158 : Where information is lacking, are donors more or less "dangerous" (carriers of the virus) in terms of transfusion?
Line 167 : How do you explain that 45.6% of donors (who are adults) do not know whether or not they have been vaccinated.
Table 2 : demographic profile of blood donors. The data (particularly detailed) are obviously interesting, but a table with the risk factors, in the context of this study, would have been just as interesting,... or even more.
Line 172 : Risk factors include the risk of dental procedures, but not invasive medical examinations such as gastroscopy, for example. What for? An oversight?
Line 177 : The significant value of p (p = 0.011) should appear in Figure 1 (not just in the text). It is understood that the variation of the HBs antigen in the 2 samples is significantly different, but the way in which it has evolved over time in the 3 categories is not obvious and deserves some explanation.
Line 184 : Figure 2. There was no significant difference in the presence/absence of anti-HBc between the 1st sample and the return, but the distribution of the results was statistically different. We would like to know more about this difference. The IgG or IgM component of anti-HBc is not known.
Line 191 : The strong correlation between the presence of HBs antigen and DNA is obviously not surprising. DNA is detected in 75% of HBs Ag-positive donors. Are the remaining 25% recovering ? or undergoing treatment ? or other?
Line 198 : Figure 3. The 3 categories "reactive", "non-reactive", "inconclusive" are defined in relation to the tests that were used. For discordant samples, why didn't you use another test? The fact of using tests of different commercial origin, or of a different basic principle, allows, in an interesting proportion of cases, to remove ambiguities.
Line 201 : The following point is entitled “3.3. Serological and infectious profiles without return” . but one can read “Based on the combination of serological and molecular markers (HBsAg, total anti-202 HBc, anti-HBs, and HBV DNA) obtained at the moment of return, it was possible to classify ….”. Isn't there a contradiction between the title and this sentence?
Line 218 : 17.4% of false positive reactions is a lot! One may wonder what the analysis of these same samples would have yielded with another test?
Line 220 : Table 3. Distribution of profiles. The left-hand column of the table should – advantageously – be divided into 4 sub-columns whose headings would be: "HBsAg"; "Anti-HBc"; "Anti-HBs" and "DNA". And the different cases indicated by "pos" or "neg" or "ind" in the corresponding lines. This important table would gain in both clarity and readability.
Line 232 : the strong association between drug use and the presence of the HBs antigen is not surprising. Since the series is made up of donors (or candidate donors), and even if the subject of the study is centered on hepatitis B, one may wonder what the results of the anti-HIV research in the same category were?
Line 235 : a word is missing in the following sentence: … « positivity was statistically significant in association with donation…”.
Line 235 : For the reader, it is a bit "confusing" to put in the same sentence that there is a correlation with the "frequency of donations" in "first-time" donors. It would be useful to rephrase that little paragraph. In other words, talking about "donation frequency" among donors who are donating for the first time is confusing. Wouldn't it be clearer to speak of "categories" or "types" of donors and calling some: "regular donors" or "former donors" ("repeate donor"), and others, first-time donors, (sometimes also called "candidate donors")?
Line 239 : indeed, a value of p (p = 0.05739). worth mentioning.
Line 240 : the increase in the frequency of anti-HBc with age testifies to an accumulation over time of occasions of contact with the HBV virus. What is more surprising is the decrease in this frequency in the category of older donors. Do you have an explanation?
Line 260 : same remark as for line 235.
Line 263 : Have you looked at whether or not there is a correlation between gender (Male, Female) and illicit drug use?
Line 271 : for the 2 very special cases WP and OBI, it would have been interesting to follow up and analyze a sample subsequent to that of the "return". How did you rule out the hypothesis of a contamination by DNA for the WP case, for example. The presence of anti-HBs is surprising, and the donor does not know if he has been vaccinated. The OBI donor would have been vaccinated ? In addition, the notion of "replacement" donor is not explained in the text and is not part of the categories of donors discussed in the study.
Discussion
Line 282 : Is there a particular reason for choosing to compare the figures to those of Ethiopia or Somalia, countries far from Brazil??. 235)
Line 288 : Caution in the use of the notion of "donation frequency". (cfr. Comment Line 235).
Line 298 – 300 : Only 1.9% of donors show a profile compatible with vaccination. We can seriously question whether the vaccination against hepatitis B is done "according to the rules", i.e. in compliance with a schedule with booster injection to guarantee its effectiveness. The authors rightly raise the question of the need for a proper vaccination policy in an endemic area such as the one in question here.
Line 320 : As far as "Health education strategies" are concerned. In this study, only once is a reference to a questionnaire to try to locate the origin of a possible contamination by the HBV virus (line 90). In most industrialized European countries, before donation, a questionnaire is submitted to the donor (who must read, complete and sign it) in order to assist in medical selection by asking donors about possible risk factors. This practice does not seem to be routinely applied in Brazil.
Line 324 : even if there are more men than women, is there a difference between the sexes when it comes to the carriage of hepatitis B markers?
Line 329 : as already reported, anti-HBc suspected of being False Positives could perhaps have been "verified" using a second method. While the increase in the frequency of these anti-HBc antibodies with age is an expected observation, the decrease observed in donors over 60 years of age is quite surprising and has apparently not been explained.
Line 363 : “high frequency of possible False Positive profiles (17.6%)". It is also possible that the positivity threshold will need to be adapted to the population studied. This can sometimes be the case, for example, when the immunoglobulin concentration of the sera tested is high compared to that of the population used by the manufacturer to validate the test.
Line 379 : There are indeed several limitations to this study, in particular the lack of follow-up over time of the disputed cases, but the call for particular attention to improve screening, vaccination and transfusion safety is very important.
Line 384-… : the series studied by the authors shows how complex the screening of blood donors for hepatitis B is and how this complexity is potentially dangerous for blood safety. While the implementation of molecular biology methods (DNA evidence) undoubtedly brings a significant advantage, the use of algorithms based on the use of several different tests can also provide valuable and less expensive help. It is nevertheless true that transfusion safety does not depend only on technical performance, but also, and probably above all, on a strategy for informing donors about risk factors, information that can enlighten and encourage a donor to self-exclude from donating blood.
Author Response
Dear Reviewer, we appreciate your attention to the evaluation of the manuscript. We greatly value the observations and suggestions presented, which significantly contributed to improving the scientific quality and clarity of the study. Below, we present the responses provided to each comment.
Comment 1: The first question that comes to mind is: "Why publish results in 2025 from samples that date back to 2015?" Why not study more recent samples? Wouldn't studying the current epidemiological profile have provided more up-to-date and useful information, especially for donor selection?
Response 1: The time lag between data collection and publication was due to time and budget constraints. However, epidemiological data obtained at different times are essential to understanding the dynamics of hepatitis B infections among blood donors. We hope that the 2015 dataset will serve as a historical reference for future comparative analyses.
Comment 2: On a technical level, one may ask: why not have used separate IgG and IgM tests for anti-HBc? Why was there no 2nd method (an alternative test) to analyze the "dubtful" samples? How are the results of the analyses validated?
Response 2: Serological screening followed the official algorithm in force in Brazil (Ministry of Health, Consolidation Ordinance No. 5, 2017). Anti-HBc was tested by detecting total antibodies (IgG + IgM). Doubtful results were repeated, and persistently reactive samples were sent for confirmatory testing (Artus HBV QS-RGQ, QIAGEN, Hilden, Germany), in accordance with national transfusion safety standards.
Comment 3: Table 2 is very detailed in relation to its informative contribution;
Response 3: Table 2 (now table 3) has been revised and simplified to make it more objective and informatively concise. The categories of the variable “Usual Occupation” have been adjusted, as well as the corresponding text (lines 160–165).
Comment 4: Figures 1 and 2 would deserve some further explanation;
Response 4: Figures 1 and 2 have been revised and given more detailed descriptions. In Figure 1, the legend has been expanded to clarify the interpretation of HBsAg (RLU/CO) values and the statistical difference between the donation and return times, and the p-value has been included in the figure itself (lines 189-190). In Figure 2, both the manuscript text (lines 192–195) and the legend have been adjusted to more clearly explain the variations observed in anti-HBc results between the two time points.
Comment 5: Figure 3 could be summarized;
Response 5: Figure 3 has been reorganized to improve readability and simplify the visualization of the individual reactivity of the markers (anti-HBc, HBsAg, and NAT). In addition, we have included a more detailed description in the manuscript text (lines 206–211) to better clarify the purpose and interpretation of the figure.
Comment 6: the presentation of Table 3 would benefit from being simplified (see comment related to Line 220).
Response 6: Thank you for the suggestion. The table (now table 5) has been reorganized as recommended (line 243).
Comment 7: Figure 4: while (we know that) the evolution of anti-HBc with age, here between [18 and 59] years of age, is increasing, we do not understand the decrease in anti-HBc observed beyond the age of 60 (too few observations? n=6).
Response 7: The reduction in anti-HBc frequency after age 60 was explained in the comments regarding lines 240 and 329. In summary, this variation is mainly due to the very small number of donors in this age group (n = 6), which may have influenced the prevalence estimate.
Comment 8: The use of "professional jargon" can be dangerous. See explanations below, in the comments on line 235 and line 260. In summary: for the reader, it is a bit "confusing" to read in the same sentence that there is a correlation between one given parameter (whatever it is) with the "frequency of donations" in "first-time" donors.
Response 8: The text has been revised to make the wording clearer. Specifically, we replaced the term "frequency of donations" with "donor type" (first donation vs. repeat donor), avoiding the possible misinterpretation mentioned.
Comment 9: Line 85: “During this period, 1,086 donors presented…” Why not put directly here the percentage that this represents, that is to say: 1.156%. This interesting notion appears only in line 278 in the discussion.
Response 9: The corresponding percentage (1.156%) was included directly in line 84, as recommended, to make the information clearer.
Comment 10: Line 120 : It is not known how the test result is "validated". Is there an algorithm that recommends starting the analysis 2 or 3 times? or who recommends using another test to validate a weakly reactive or inconclusive result, ...
Response 10: Inconclusive or weakly reactive samples were retested three consecutive times with the same assay. Samples that remained reactive were subjected to an alternative confirmatory test (Artus HBV QS-RGQ, QIAGEN, Hilden, Germany). This two-step approach ensures analytical reliability and reduces the occurrence of false positives. The explanation was added in lines 112-116.
Comment 11: Line 151 : How long did it take between the first and second sampling? Was this delay more or less the same for all retested donors?
Response 11: The interval between the first and second collection ranged from 7 to 389 days (mean: 78.6; median: 37.5). Most donors (69.5%) returned within three months of the first donation. This information was added to the Results (lines 155–157).
Comment 12: Line 154 : The age range for donors is 16 to 63 years. Is 63 years old the maximum age allowed to donate blood? (in other countries, it's 65 years old).
Response 12: The number 63 does not refer to the maximum age for blood donation. This number corresponds to the maximum age reported within the sample included in the study. The maximum age for blood donation in Brazil is 65 years.
Comment 13: Line 157 : The majority of donors in the series studied are "first time donors" (76%). In some countries, at the time of AIDS in the 1980s and later in the 1990s, when hepatitis C tests were available, there was a rather peculiar phenomenon: people came forward to donate blood, but in reality, their motivation for going to the blood bank was to get tested. Could the same phenomenon be one of the possible explanations ? Or, is the frequency of hepatitis B in the general population so important?
Response 13: Determining whether donors seek testing as motivation would require a different research design, with a qualitative approach and a focus on donation intention. This study was based solely on secondary data. The high proportion of first-time donors may be related to the nature of blood donation campaigns, which are conducted focusing on different regions of the city, leading to mass dissemination that generates many first-time donations. The literature already has clear data on the endemic nature of hepatitis B in the Brazilian Amazon, including the general population, even though we have a vaccine.
Comment 14: Line 157 : Have you had access to the serological results (Test Ratio) characterizing the first donation?
Response 14: The numerical values of the reactivity index (Test Ratio) of the first donation were not fully available in the original records. However, according to the technical report used at the time, the criterion adopted for the selection of samples considered those with an index equal to or greater than 1.2 as positive.
Comment 15: Line 158 : Where information is lacking, are donors more or less "dangerous" (carriers of the virus) in terms of transfusion?
Response 15: The nine donors who did not report the type of donation (3.5% of the sample) were evaluated individually. Most presented a serological profile compatible with a resolved previous infection (reactive anti-HBc, reactive anti-HBs, negative HBsAg and NAT), while the others were classified as vaccinated, susceptible, or possible false positives. Importantly, none of them presented markers of active infection (detectable HBsAg or DNA). Therefore, this subgroup does not represent a higher transfusion risk compared to the other participants. Furthermore, even in the absence of information on donation history, all collected blood bags undergo mandatory serological and molecular testing before release. Any reactive result leads to the immediate discarding of the bag, ensuring transfusion safety. As this group is small and does not alter the interpretation of the findings, we chose not to modify the manuscript text.
Comment 16: Line 167 : How do you explain that 45.6% of donors (who are adults) do not know whether or not they have been vaccinated.
Response 16: The high proportion of donors who were unable to report their vaccination status likely reflects the common difficulty adults have in recalling their vaccination history. Many individuals may have been vaccinated many years ago, especially in mass vaccination campaigns or school settings, without individual records or with lost vaccination cards. Furthermore, some participants may confuse the hepatitis B vaccine with other vaccines received throughout their lives. Thus, the "don't know" response primarily reflects limitations in memory or documentation, and not necessarily an absence of vaccination.
Comment 17: Table 2 : demographic profile of blood donors. The data (particularly detailed) are obviously interesting, but a table with the risk factors, in the context of this study, would have been just as interesting,... or even more.
Response 17: In response to the comment, we have included in the manuscript a new table containing the main behavioral and clinical factors associated with the risk of HBV infection (Table 4, line 177).
Comment 18: Line 172 : Risk factors include the risk of dental procedures, but not invasive medical examinations such as gastroscopy, for example. What for? An oversight?
Response 18: The risk factors were based on the official donor screening questionnaire (Consolidation Ordinance No. 5, 2017). Dental procedures are explicitly listed, while invasive examinations, such as gastroscopy, are included in the category 'medical procedures with contact with blood or biological material'. This was not an oversight, but rather the format for collecting the information.
Comment 19: Line 177 : The significant value of p (p = 0.011) should appear in Figure 1 (not just in the text). It is understood that the variation of the HBs antigen in the 2 samples is significantly different, but the way in which it has evolved over time in the 3 categories is not obvious and deserves some explanation.
Response 19: The p-value (p = 0.01173) was included directly in Figure 1, as requested. Furthermore, we added a detailed explanation to the text regarding the evolution of the three HBsAg categories between the time of donation and the return visit, since the variation was not evident from the figure alone. Specifically, we described that: most donors remained non-reactive in both samples; one initially reactive individual became non-reactive upon return; one initially inconclusive donor showed a reactive result in the second evaluation. These individual transitions explain the statistical difference observed between the two time points. This description was included in the Results (lines 182-187).
Comment 20: Line 184 : Figure 2. There was no significant difference in the presence/absence of anti-HBc between the 1st sample and the return, but the distribution of the results was statistically different. We would like to know more about this difference. The IgG or IgM component of anti-HBc is not known.
Response 20: In fact, the overall proportion of donors positive for anti-HBc remained practically stable between the two collections. However, the analysis showed small individual variations: some initially reactive donors presented inconclusive or non-reactive results upon return, while others exhibited the opposite pattern. These subtle changes explain the observed statistical difference (p = 0.0089) and likely reflect transient variations in the immune response or analytical fluctuations (lines 192-195). We also emphasize that the assay used detects total anti-HBc, without distinguishing between IgM and IgG components, which limits inferences about the stage of infection.
Comment 21: Line 191 : The strong correlation between the presence of HBs antigen and DNA is obviously not surprising. DNA is detected in 75% of HBs Ag-positive donors. Are the remaining 25% recovering ? or undergoing treatment ? or other?
Response 21: Regarding the 25% of HBsAg-positive donors who did not show detectable HBV DNA, we found that these three individuals had HBsAg values very close to the assay cutoff, ranging from 1.57 to 7.00 RLU/CO. At both testing time points, they remained non-reactive for anti-HBc and anti-HBs and continued to test negative for HBV DNA. This pattern, characterized by isolated low-level HBsAg reactivity and the absence of any additional serological or molecular markers, strongly supports the interpretation of false-positive HBsAg results, which is a known phenomenon in samples with borderline reactivity. We acknowledge that other explanations may be considered at a purely theoretical level, such as the presence of viral variants with mutations in genomic regions targeted by the molecular assay. However, these possibilities are not supported by the available data and should be regarded only as speculative. Therefore, the most consistent explanation for the absence of detectable HBV DNA in these cases is analytical variation of the immunoassay rather than a biological process, such as recovery from infection or ongoing treatment.
Comment 22: Line 198 : Figure 3. The 3 categories "reactive", "non-reactive", "inconclusive" are defined in relation to the tests that were used. For discordant samples, why didn't you use another test? The fact of using tests of different commercial origin, or of a different basic principle, allows, in an interesting proportion of cases, to remove ambiguities.
Response 22: The analyses followed the official algorithm of the Ministry of Health (Consolidation Ordinance No. 5/2017). Initially reactive or inconclusive samples were repeated in the same assay, and persistently reactive samples were submitted to the confirmatory molecular test (Artus HBV QS-RGQ, QIAGEN, Hilden, Germany), based on real-time PCR. Therefore, there was no need to use kits from other manufacturers, as the samples were repeated and confirmed by a method with a different principle.
Comment 23: Line 201 : The following point is entitled “3.3. Serological and infectious profiles without return” . but one can read “Based on the combination of serological and molecular markers (HBsAg, total anti-202 HBc, anti-HBs, and HBV DNA) obtained at the moment of return, it was possible to classify ….”. Isn't there a contradiction between the title and this sentence?
Response 23: We thank the reviewer for the observation. The section title has been adjusted to “Serological and infectious profiles at return” to accurately reflect that the data presented correspond to donors who returned for follow-up collection.
Comment 24: Line 218 : 17.4% of false positive reactions is a lot! One may wonder what the analysis of these same samples would have yielded with another test?
Response 24: In fact, the 17.4% false-positive rate may seem high, but it is within the expected range for screening tests used in blood banks, whose purpose is to ensure high sensitivity, even at the expense of specificity, aiming for maximum transfusion safety. These results are inherent to immunological screening tests, especially in areas of high endemicity, such as the Amazon region. We emphasize that samples with inconclusive or reactive results were subsequently submitted to retesting and/or molecular confirmation (PCR for HBV), according to the institutional protocol.
Comment 25: Line 220 : Table 3. Distribution of profiles. The left-hand column of the table should – advantageously – be divided into 4 sub-columns whose headings would be: "HBsAg"; "Anti-HBc"; "Anti-HBs" and "DNA". And the different cases indicated by "pos" or "neg" or "ind" in the corresponding lines. This important table would gain in both clarity and readability.
Response 25: Thank you for your suggestion. Table 3 (now Table 5) has been revised as recommended (line 243).
Comment 26: Line 232 : the strong association between drug use and the presence of the HBs antigen is not surprising. Since the series is made up of donors (or candidate donors), and even if the subject of the study is centered on hepatitis B, one may wonder what the results of the anti-HIV research in the same category were?
Response 26: The database used in this study was built exclusively with results related to HBV markers, not including information on HIV. Although we recognize the relevance of this comparative analysis, it could not be performed due to the absence of this data in the analyzed set. To investigate the association between variables and the risk of HIV infection, it would be necessary to conduct a specific epidemiological study focused on this topic.
Comment 27: Line 235 : a word is missing in the following sentence: … « positivity was statistically significant in association with donation…”.
Response 27: The correction has been made.
Comment 28: Line 235 : For the reader, it is a bit "confusing" to put in the same sentence that there is a correlation with the "frequency of donations" in "first-time" donors. It would be useful to rephrase that little paragraph. In other words, talking about "donation frequency" among donors who are donating for the first time is confusing. Wouldn't it be clearer to speak of "categories" or "types" of donors and calling some: "regular donors" or "former donors" ("repeate donor"), and others, first-time donors, (sometimes also called "candidate donors")?
Response 28: We thank the reviewer for the comment. The sentence has been revised to eliminate the ambiguity between ‘donation frequency’ and ‘first-time donors’. The term has been replaced by ‘donor type’, clearly distinguishing between ‘first-time donors’ and ‘repeat donors’. The corresponding changes can be found in lines 260 to 262 of the manuscript.
Comment 29: Line 239 : indeed, a value of p (p = 0.05739). worth mentioning.
Response 29: The p-value of 0.057 was retained in the text and the finding was highlighted as a trend that may indicate a possible behavioral influence, lines 265-266.
Comment 30: Line 240 : the increase in the frequency of anti-HBc with age testifies to an accumulation over time of occasions of contact with the HBV virus. What is more surprising is the decrease in this frequency in the category of older donors. Do you have an explanation?
Response 30: The text has been adjusted to acknowledge the slight reduction in anti-HBc positivity among donors over 60 years of age. This variation may be related to the small number of participants in this age group (n = 6), which may have influenced the prevalence estimate. Lines 269-271.
Comment 31: Line 260 : same remark as for line 235.
Response 31: The correction has been made.
Comment 32: Line 263 : Have you looked at whether or not there is a correlation between gender (Male, Female) and illicit drug use?
Response 32: The additional analysis evaluating the association between gender and illicit drug use revealed a statistically significant relationship (p = 0.0003), with an estimated odds ratio of 6.84 (95% CI: 1.99–36.50), indicating a higher likelihood of illicit drug use among male donors. This information has been incorporated into the manuscript (Section 3.5, lines 296–299).
Comment 33: Line 271 : for the 2 very special cases WP and OBI, it would have been interesting to follow up and analyze a sample subsequent to that of the "return". How did you rule out the hypothesis of a contamination by DNA for the WP case, for example. The presence of anti-HBs is surprising, and the donor does not know if he has been vaccinated. The OBI donor would have been vaccinated ? In addition, the notion of "replacement" donor is not explained in the text and is not part of the categories of donors discussed in the study.
Response 33: The study was planned to include only two collections, the first donation and the return sample, without subsequent follow-up. Therefore, it was not possible to perform a new collection to monitor the cases classified as WP and OBI. In the case identified as WP, we observed the presence of HBV DNA, even without detectable serological markers. The possibility of laboratory contamination was considered very low, since all internal controls were adequate. The presence of reactive anti-HBs may indicate a very recent seroconversion, when antibodies are beginning to appear, or just a borderline result, without a direct relationship to previous vaccination. The donor classified as OBI presented reactive anti-HBc and detectable DNA, but a negative result for HBsAg. This participant reported having received the HBV vaccine, which may represent an incomplete vaccine response or an old infection, with persistence of very low levels of viral DNA, as occurs in occult infections. Finally, the term "replacement donor" is an administrative classification within the blood transfusion service, used to identify donors who donate on behalf of a specific patient. Since this variable was not used in the comparative analyses, we decided to remove it from the revised text to avoid confusion.
Comment 34: Line 282 : Is there a particular reason for choosing to compare the figures to those of Ethiopia or Somalia, countries far from Brazil??. 235)
Response 34: Comparisons with data from Ethiopia and Somalia were included for contextual reference, since these countries have historically high rates of serological markers for HBV and are widely cited in global prevalence studies. The aim was to show that, although the rate observed in the present study is comparable to that of other Brazilian regions, it remains substantially lower than that recorded in areas of high endemicity, such as certain African countries.
Comment 35: Line 288 : Caution in the use of the notion of "donation frequency". (cfr. Comment Line 235).
Response 35: The correction has been made.
Comment 36: Line 298 – 300 : Only 1.9% of donors show a profile compatible with vaccination. We can seriously question whether the vaccination against hepatitis B is done "according to the rules", i.e. in compliance with a schedule with booster injection to guarantee its effectiveness. The authors rightly raise the question of the need for a proper vaccination policy in an endemic area such as the one in question here.
Response 36: Indeed, the low proportion of donors with a profile compatible with vaccine immunity (1.9%) reinforces the need for attention to HBV vaccination strategies. This finding may reflect both failures in complete adherence to the three-dose schedule and the absence of booster campaigns targeting the adult population, especially in endemic regions. We chose to continue the discussion in this vein, highlighting that the data suggest possible gaps in vaccination coverage and effectiveness, and reinforce the importance of public policies aimed at adequate immunization of susceptible adults.
Comment 37: Line 320 : As far as "Health education strategies" are concerned. In this study, only once is a reference to a questionnaire to try to locate the origin of a possible contamination by the HBV virus (line 90). In most industrialized European countries, before donation, a questionnaire is submitted to the donor (who must read, complete and sign it) in order to assist in medical selection by asking donors about possible risk factors. This practice does not seem to be routinely applied in Brazil.
Response 37: The mention of these actions was proactive, aiming to highlight the importance of developing programs focused on groups in more vulnerable situations, such as users of illicit drugs. In Brazil, all blood donation candidates undergo an individual clinical screening before donation, conducted by professionals from the blood center, in which the donor orally answers questions about possible risk behaviors and receives guidance on safe practices and infection prevention. Even so, there is a lack of continuous and specific educational strategies aimed at key populations outside the donation environment. Therefore, in lines 352–355, we sought to improve the explanation, making it clearer that the reference to health education strategies is proactive and complements the screening already carried out in the country. Thus, we reinforce the importance of strengthening public policies and awareness campaigns, especially among groups exposed to risk behaviors, as an essential part of the efforts to prevent and control hepatitis B.
Comment 38: Line 324 : even if there are more men than women, is there a difference between the sexes when it comes to the carriage of hepatitis B markers?
Response 38: Thank you for your observation. We performed separate exploratory analyses between gender and each hepatitis B marker. The only statistically significant association was observed for HBV DNA, which is more frequent among men, possibly reflecting greater viral exposure in this group. Although the sample contains more men than women (147 vs. 112), the proportion of individuals with at least one reactive serological marker was practically identical between the sexes: 83.9% among women and 83.7% among men. Fisher's exact test confirmed the absence of a statistically significant difference (p = 1.0; OR = 0.98; 95% CI: 0.47–2.01). This analysis was conducted in a complementary manner, solely to clarify the question raised, and was not included in the manuscript as it exceeded the original scope of the descriptive study. Finally, we reformulated lines 357–360 to make them clearer and more aligned with the findings.
Comment 39: Line 329 : as already reported, anti-HBc suspected of being False Positives could perhaps have been "verified" using a second method. While the increase in the frequency of these anti-HBc antibodies with age is an expected observation, the decrease observed in donors over 60 years of age is quite surprising and has apparently not been explained.
Response 39: We acknowledge that confirming suspected false-positive anti-HBc results using a second method could have increased the robustness of the findings. However, we emphasize that inconclusive or weakly reactive samples were retested three consecutive times with the same assay, and those that remained reactive were subjected to an alternative confirmatory test (Artus HBV QS-RGQ, QIAGEN, Hilden, Germany). Regarding the variation in anti-HBc positivity among age groups, the expected increasing trend with advancing age was observed, reflecting the accumulation of exposures throughout life. The slight reduction observed among donors over 60 years of age is probably due to the small number of participants in this group (n = 6), which may have influenced the stability of the estimates. For greater clarity, we highlight this observation about the sample size in lines 367-369 of the discussion.
Comment 40: Line 363 : “high frequency of possible False Positive profiles (17.6%)". It is also possible that the positivity threshold will need to be adapted to the population studied. This can sometimes be the case, for example, when the immunoglobulin concentration of the sera tested is high compared to that of the population used by the manufacturer to validate the test.
Response 40: We recognize that individual differences, such as variations in immunoglobulin concentration in the studied population, can influence test performance and contribute to false-positive results. We hope to be able to access this type of information in future studies.
Comment 41: Line 379 : There are indeed several limitations to this study, in particular the lack of follow-up over time of the disputed cases, but the call for particular attention to improve screening, vaccination and transfusion safety is very important.
Response 41: In fact, we identified as a limitation the absence of longitudinal follow-up of inconclusive cases, which would have allowed for better confirmation of the results and understanding of the serological evolution. Even so, we emphasize that the findings reinforce the importance of continuously improving screening strategies, expanding vaccination coverage, and strengthening actions aimed at transfusion safety.
Comment 42: Line 384-… : the series studied by the authors shows how complex the screening of blood donors for hepatitis B is and how this complexity is potentially dangerous for blood safety. While the implementation of molecular biology methods (DNA evidence) undoubtedly brings a significant advantage, the use of algorithms based on the use of several different tests can also provide valuable and less expensive help. It is nevertheless true that transfusion safety does not depend only on technical performance, but also, and probably above all, on a strategy for informing donors about risk factors, information that can enlighten and encourage a donor to self-exclude from donating blood.
Response 42: We appreciate the comment and agree with the observation. Our results truly highlight the complexity of hepatitis B screening and show that, in addition to molecular methods, algorithms that integrate different markers can offer important support for transfusion safety. We also recognize that this safety depends not only on testing, but also on good donor education regarding risk factors and the importance of self-exclusion when necessary.
Reviewer 2 Report
Comments and Suggestions for Authors
In the manuscript entitled "Epidemiological Profile and Diagnostic Outcomes of Blood Donors Following Hepatitis B Screening at the Largest Blood Bank in the State of Pará, Brazil" the authors perform serological and molecular screening on blood samples from donors that have been deemed ineligible for blood donation. The authors collect demographic information and attempt to corelate demographic data with the presence of HBV. This study provides important information regarding HBV prevalence and risks in HBV endemic regions. Please revise the manuscript as below.
1) In lines 85-86, can the authors clarify the laboratory screening used for HBV by the hospitals? Since the authors mention that NAT is a more robust test for screening, is that included in the screening performed by the hospitals? Additionally, if a donor presents a profile of resolved HBV infection, HBV-reactive Abs present but HBV antigens absent, does that still qualify the patient as ineligible as a blood donor?
2) The window period has been defined differently in the methods and in table 3. Please rectify table 3.
3) Section 2.4 can be prepared as a table for the benefit of the readers.
4) Out of the 32% donors that reported having received the HBV vaccination, were there any positive for infections or resolved infections? In other words, does the data hint towards reduced immune responses after a period of vaccination and is it necessary to perform re-vaccinations in regions that are endemic for HBV?
5) The individuals that had active infection at the time of donation were the same 12 individuals that were positive again upon return. And 11 of which were positive for HBV DNA? Please add this information to the description.
6) Minor point- moment should be referred to as time points.
7) The number of people that have been regarded as positive false cases and there are several individuals, do the authors think that these individuals have resolved infections but with low levels of circulating antibodies? It is very unlikely that 45 individuals will have inconclusive or low detection levels of both anti-HBVs and anti-HBVc simultaneously.
8) Illicit drug use has been associated with other high-risk behaviors without providing any citations in the discussion section. lines 313-315.
9) The authors mention anti-HBc isolated group. Is it known from the literature that the humoral responses to HBs are not as durable as the ones to HBc. Could that be the reason for this group?
10) Lastly, in the 6.6 % donors that were listed as susceptible to HBV, were there any vaccinated individuals?
Author Response
Dear Reviewer, we appreciate your attention to the evaluation of the manuscript. We greatly value the observations and suggestions presented, which significantly contributed to improving the scientific quality and clarity of the study. Below, we present the responses provided to each comment.
Comment 1: In lines 85-86, can the authors clarify the laboratory screening used for HBV by the hospitals? Since the authors mention that NAT is a more robust test for screening, is that included in the screening performed by the hospitals? Additionally, if a donor presents a profile of resolved HBV infection, HBV-reactive Abs present but HBV antigens absent, does that still qualify the patient as ineligible as a blood donor?
Response 1: At the blood transfusion service where the study was conducted (HEMOPA Foundation), laboratory screening for hepatitis B follows the guidelines of the Brazilian Ministry of Health. All donors are tested for HBsAg, total anti-HBc, and NAT-HBV. The NAT test (Nucleic Acid Test) is an integral part of the routine screening protocol, performed on pooled samples, with the aim of detecting the presence of viral DNA early and reducing the risk of the immunological window. Regarding eligibility, according to Brazilian serological screening standards, donors with a profile compatible with resolved infection (reactive anti-HBc and reactive anti-HBs, with non-reactive HBsAg and undetectable DNA) are considered permanently ineligible for blood donation due to a history of HBV exposure. This measure aims to ensure the highest level of transfusion safety, even in the absence of active infection.
Comment 2: The window period has been defined differently in the methods and in table 3. Please rectify table 3.
Response 2: The correction has been made.
Comment 3: Section 2.4 can be prepared as a table for the benefit of the readers.
Response 3: The correction has been made.
Comment 4: Out of the 32% donors that reported having received the HBV vaccination, were there any positive for infections or resolved infections? In other words, does the data hint towards reduced immune responses after a period of vaccination and is it necessary to perform re-vaccinations in regions that are endemic for HBV?
Response 4: Among the 83 donors (32%) who reported receiving the HBV vaccine, the majority (approximately 60%) presented a profile compatible with a previously resolved infection (reactive anti-HBc and anti-HBs, with undetectable HBsAg and DNA). A smaller number of donors presented a susceptible profile (about 4%), isolated anti-HBc (1%), or inconclusive/false positive results (approximately 5%). In addition, five donors presented a profile compatible with vaccine-induced immunity, with reactive anti-HBs and absence of other markers, with no detection of viral DNA. Among these individuals, three did not know if they had been vaccinated, one confirmed receiving the vaccine, and one denied vaccination. Despite these discrepancies in the responses, all exhibited the same serological pattern compatible with vaccine-induced immunity, reinforcing the consistency of the laboratory results. These findings indicate that, even among vaccinated individuals or those with a previous vaccine response, a progressive decline in protective antibody levels (anti-HBs) or failures in the immune response may occur, highlighting the importance of monitoring immunity and considering vaccine booster strategies in endemic regions, such as the Amazon. We seek to clarify this issue in the paragraph on lines 332–336.
Comment 5: The individuals that had active infection at the time of donation were the same 12 individuals that were positive again upon return. And 11 of which were positive for HBV DNA? Please add this information to the description.
Response 5: In the present study, the classification of participants regarding HBV infection status was performed based on the results obtained after the return visit, considering the set of serological and molecular markers. Nine donors were classified as having an active infection, corresponding to 3.5% (9/259) of the sample. All showed positivity for HBsAg and detectable DNA at the time of the return visit. These same individuals had already shown positivity for HBsAg at the time of donation, which was confirmed in the follow-up samples. Among them, eight presented detectable HBV DNA in both the donation and return samples, while one case did not have the molecular test performed on the first sample, but maintained a reactive HBsAg result in both collections. These findings confirm the persistence of the infection and the consistency of the serological and molecular profiles throughout the follow-up. The text in the Results section has been adjusted to incorporate this information (lines 226–233), as requested. Furthermore, we have made it clear in the methodology that this classification took into account the results obtained at the time of return.
Comment 6: Minor point- moment should be referred to as time points.
Response 6: The correction has been made.
Comment 7: The number of people that have been regarded as positive false cases and there are several individuals, do the authors think that these individuals have resolved infections but with low levels of circulating antibodies? It is very unlikely that 45 individuals will have inconclusive or low detection levels of both anti-HBVs and anti-HBVc simultaneously.
Response 7: Some individuals classified as “possible false positives” presented serological patterns suggestive of weak or inconsistent reactivity, especially for anti-HBc and/or anti-HBs, with undetectable results for HBV DNA. It is plausible that some of these cases may indicate a previous resolved infection, with very low levels of residual antibodies, close to the lower limit of detection of the assays used. However, in the absence of additional quantitative tests or confirmatory re-collection, it was not possible to definitively distinguish between residual results and nonspecific reactions. For this reason, we retained the descriptive term “possible false positives” in the manuscript, adding in the Discussion section the observation that some of these profiles may reflect old infections with minimal serum antibody levels (lines 407-411).
Comment 8: Illicit drug use has been associated with other high-risk behaviors without providing any citations in the discussion section. lines 313-315.
Response 8: A reference (Sousa et al. 2018) addressing this association has been added. The study identified, through multivariate analysis, that illicit drug use, unprotected sex, and multiple sexual partners are associated with HBV infection. This citation was included on line 348 of the Discussion section.
Comment 9: The authors mention anti-HBc isolated group. Is it known from the literature that the humoral responses to HBs are not as durable as the ones to HBc. Could that be the reason for this group?
Response 9: The 16 individuals classified as “isolated anti-HBc” showed reactive anti-HBc in both samples, with an absence of HBsAg, anti-HBs, and detectable DNA. These cases may be consistent with a previous resolved infection, with loss of anti-HBs over time, since the humoral response to anti-HBc generally persists throughout life. Although it is not possible to completely rule out the occurrence of nonspecific reactivity, this interpretation has been incorporated into the Discussion section (lines 383–390).
Comment 10: Lastly, in the 6.6 % donors that were listed as susceptible to HBV, were there any vaccinated individuals?
Response 10: Among the 17 donors classified as susceptible to HBV (6.6%), four reported prior vaccination (23.5%), eight did not know (47.1%), four stated they had not been vaccinated (23.5%), and one had no recorded information. All presented non-reactive results for the serological and molecular markers tested. These findings suggest that some individuals who consider themselves vaccinated may not have developed a detectable immune response, possibly due to incomplete schedules, vaccine failure, or loss of immunity over time. This explanation was added to the Discussion section (lines 398-402).
Round 2
Reviewer 1 Report
Comments and Suggestions for Authors
To the Authors
I very much appreciated the answers you gave to the (very) many comments I had made. There is a very clear improvement in the text both in terms of its content, but also in its presentation: there is more clarity on the points that deserved some explanation and also a significant improvement in the format of presentation of the tables.
Reading the responses to the comments was like an enjoyable and fruitful scientific conversation (this is not always the case!). Thank you also for answering the questions that deviated a little from the subject.
I will take up the points one by one in a table below, but I also propose the following remarks, which essentially concern the form.
4 Comments concerning version 02
Point 1: In this article, there are 5 tables and 4 figures; that's a lot. It is true that the tables have made it possible to greatly simplify and clarify the presentation of certain data (I am thinking specially of Table 1), but we must nevertheless try to remain as concise as possible.
Point 2 : Table 3 could be advantageously simplified (especially since you show that these demographic factors are not significantly related to the markers studied). For example, grouping the items mentioned into 2 to 4 categories maximum: for "usual occupation"; for "school level", and "marital status". This will decrease the number of lines.
Point 3 : On the other hand, Table 4: “risk factors” which is very interesting, you could try putting as column headers, for example: Yes, Always, Sometimes, No, Never, Ignored, which would bring the number of rows down to 8 instead of 24.
Point 4 : Line 297-299: In the last part of point 3.5. The sentence “The same pattern of non-significance was observed for the remaining variables analyzed” should be placed before that which begins with “Additionally”.
The following Table summarizes the few remaining remarks:
|
1 |
OK |
|
2 |
OK (line 113) This response (concerning both technical tests and algorithm) should be added in 2.2. and/or 2.4. |
|
3 |
Now Table 3 : OK (see Point 2 below) |
|
4 |
Figure 2 : the p-value is not indicated on the graph (p = 0,0089). (like for Fig.1) |
|
5 |
Anti_HBc should be written “anti-HBc” In figure 3 : “NAT” or “HBV DNA” ? |
|
6 |
Line 221: you can now add “refer to Table 1. |
|
7 |
Fig 4: anti-HBc decline in the 60+ age class….To demonstrate this, or to be sure of it, it would be necessary to calculate the mean and standard deviation of each point. You could also put the segment that connects 46-59 to 60+ in dashed lines subject to the small number of samples in the last class, |
|
8 |
Ok |
|
9 |
Ok |
|
10 |
Ok |
|
11 |
Ok |
|
12 |
Ok |
|
13 |
Ok |
|
14 |
Wouldn't it be interesting to point out in Material & Method that the TRs of the samples were = or > 1.2. ? |
|
15 |
Ok |
|
16 |
Ok |
|
17 |
It's a really nice idea to have added this table 4, but see Point 4 (a formal suggestion). |
|
18 |
… and this explains it. |
|
19 |
Figure 2: the p-value should be indicated in the graph like in fig 1. (p = 0,0089) |
|
20 |
Ok |
|
21 |
OK, thank you for the very clear explanation |
|
22 |
Ok |
|
23 |
Ok |
|
24 |
Ok (FP due to high IgG level ?) |
|
25 |
Ok |
|
26 |
Ok (indeed; another paper perhaps ?) |
|
27 |
Ok |
|
28 |
Ok |
|
29 |
Ok |
|
30 |
OK See also comment N°7 above in this table. An alternative would perhaps be to put the line that separates the value (46-59) to the value (60+) in dotted lines (?). |
|
31 |
Ok |
|
32 |
Ok |
|
33 |
Ok thank you for these explanations once again very clear |
|
34 |
OK, indeed |
|
35 |
Ok |
|
36 |
OK in fact, it will be very useful to remind the recommendations for vaccination against hepatitis B |
|
37 |
Ok |
|
38 |
OK and thank you, once again, for answering a question I was asking myself and which was, in fact, a little outside the objective of this study. |
|
39 |
Ok |
|
40 |
Ok |
|
41 |
This is, indeed, the most important objective. |
|
42 |
OK, we share the same opinion on this subject |
Author Response
Dear Reviewer, we sincerely appreciate the time and attention dedicated to the evaluation of our manuscript. Your detailed observations and constructive suggestions were extremely valuable and contributed substantially to enhancing both the scientific rigor and the clarity of our work. We are grateful for the careful reading and for the thoughtful feedback provided, which helped us refine important aspects of the content, presentation, and data organization. Below, we provide our point-by-point responses to each of the comments.
Table remaining remarks, with answers:
|
2 |
OK (line 113) This response (concerning both technical tests and algorithm) should be added in 2.2. and/or 2.4. |
Response: The answers regarding the algorithm were added in lines 104-108, and the explanation of the technical tests in lines 116-122 in section 2.2.
|
3 |
Now Table 3: OK (see Point 2 below) |
Response: Thank you for the suggestion. The categories for usual occupation, educational level, and marital status were simplified to a maximum of four groups each, as recommended, and the changes were incorporated into Table 3.
|
4 |
Figure 2: the p-value is not indicated on the graph (p = 0,0089). (like for Fig.1) |
Response: Thank you for the observation. The p-value (p = 0.0089) has now been included in Figure 2, as indicated in lines 210–211.
|
5 |
Anti_HBc should be written “anti-HBc” In figure 3: “NAT” or “HBV DNA” ? |
Response: Both corrections were made in Figure 3 (lines 226–230).
|
6 |
Line 221: you can now add “refer to Table 1. |
Response: The information was added to lines 254–255.
|
7 |
Fig 4: anti-HBc decline in the 60+ age class….To demonstrate this, or to be sure of it, it would be necessary to calculate the mean and standard deviation of each point. You could also put the segment that connects 46-59 to 60+ in dashed lines subject to the small number of samples in the last class, |
Response: As requested, we calculated the mean and standard deviation of age for each age group among anti-HBc reactive donors. The results confirmed the expected age distribution in each category (18–25: 21.9 ± 2.47; 26–35: 30.8 ± 2.52; 36–45: 40.3 ± 2.93; 46–59: 50.7 ± 3.20; 60+: 61.3 ± 1.51), and this information was added to the description of Figure 4 (lines 291–292). Furthermore, the segment connecting the 46–59 and 60+ groups in Figure 4 was modified to a dashed line, as suggested.
|
14 |
Wouldn't it be interesting to point out in Material & Method that the TRs of the samples were = or > 1.2. ? |
Response: The criteria related to the Test Ratio (TR) values are already highlighted in the Materials and Methods section (lines 109–115), where we describe the cut-off definition for each serological marker. In that paragraph, we specify that samples with RLU/CO values below 1.2 are considered non-reactive or inconclusive, while values ≥1.2 are classified as reactive.
|
17 |
It's a really nice idea to have added this table 4, but see Point 4 (a formal suggestion) |
Response: Thank you for the suggestion. We have updated Table 4 (line 189).
|
19 |
Figure 2: the p-value should be indicated in the graph like in fig 1. (p = 0,0089) |
Response: Thank you for the observation. The p-value (p = 0.0089) has now been included in Figure 2, as indicated in lines 210–211.
|
24 |
Ok (FP due to high IgG level ?) |
Response: Elevated IgG levels may indeed contribute to nonspecific reactivity and could help explain part of the false-positive results. Although this cannot be confirmed for each sample, this possibility is consistent with what is described in endemic regions.
|
26 |
Ok (indeed; another paper perhaps ?) |
Response: Yes, this comparison could be explored in a future study, as we are already developing research related to HIV in blood donors.
|
30 |
OK See also comment N°7 above in this table. An alternative would perhaps be to put the line that separates the value (46-59) to the value (60+) in dotted lines (? |
Response: The adjustments suggested in Comment 7 were implemented, including the use of a dashed line between the 46–59 and 60+ groups in Figure 4 (lines 289-292).
|
36 |
OK in fact, it will be very useful to remind the recommendations for vaccination against hepatitis B |
Response: We added the HBV vaccine recommendations to the discussion (lines 353-357).
Regarding the new comments, the corresponding responses are below.
Point 1: In this article, there are 5 tables and 4 figures; that's a lot. It is true that the tables have made it possible to greatly simplify and clarify the presentation of certain data (I am thinking specially of Table 1), but we must nevertheless try to remain as concise as possible.
Response 1: We understand the concern regarding the number of tables and figures. However, given the complexity of HBV serological and molecular profiles, these elements are necessary to ensure clarity and accurate interpretation of the results. Even so, we have revised the structure to improve conciseness, including simplifying Table 3, which previously contained a large amount of information.
Point 2 : Table 3 could be advantageously simplified (especially since you show that these demographic factors are not significantly related to the markers studied). For example, grouping the items mentioned into 2 to 4 categories maximum: for "usual occupation"; for "school level", and "marital status". This will decrease the number of lines.
Response 2: The categories for usual occupation, educational level, and marital status were simplified to a maximum of four groups each, as recommended, and the changes were incorporated into Table 3.
Point 3 : On the other hand, Table 4: “risk factors” which is very interesting, you could try putting as column headers, for example: Yes, Always, Sometimes, No, Never, Ignored, which would bring the number of rows down to 8 instead of 24.
Response 3: Thank you for the suggestion. We have updated Table 4.
Point 4 : Line 297-299: In the last part of point 3.5. The sentence “The same pattern of non-significance was observed for the remaining variables analyzed” should be placed before that which begins with “Additionally”.
Response 4: Thank you for the observation, we have added it to lines 314.
Furthermore, we made minor corrections to the manuscript. In Figure 3, the terms ‘detectable’ and ‘undetectable’, which were in Portuguese, were adjusted to their English versions. We also identified two inconsistencies in Table 4 and made the necessary adjustments: in the ‘No’ item of Illicit Drug Use, the percentage was updated from 87.4% to 87.7%; and in the ‘Ignored’ item of History of Blood Transfusion, from 1.2% to 1.1%. All changes were reviewed to ensure the consistency of the data presented.